# Inferior vagal ganglion galaninergic response to gastric ulcers

**Michal Zalecki**[1]*, **Judyta Juranek**[2], **Zenon Pidsudko**[1], **Marzena Mogielnicka-Brzozowska**[3], **Jerzy Kaleczyc**[1], **Amelia Franke-Radowiecka**[1]

**1** Department of Animal Anatomy, Faculty of Veterinary Medicine, University of Warmia and Mazury, Olsztyn, Poland, **2** Department of Human Physiology and Pathophysiology, School of Medicine, University of Warmia and Mazury, Olsztyn, Poland, **3** Department of Animal Biochemistry and Biotechnology, Faculty of Animal Bioengineering, University of Warmia and Mazury, Olsztyn, Poland

* michal.zalecki@uwm.edu.pl

**Data Availability Statement:** All relevant data are within the manuscript and its Supporting information files.

## Abstract

Galanin is a neuropeptide widely expressed in central and peripheral nerves and is known to be engaged in neuronal responses to pathological changes. Stomach ulcerations are one of the most common gastrointestinal disorders. Impaired stomach function in peptic ulcer disease suggests changes in autonomic nerve reflexes controlled by the inferior vagal ganglion, resulting in stomach dysfunction. In this paper, changes in the galaninergic response of inferior vagal neurons to gastric ulceration in a pig model of the disease were analyzed based on the authors' previous studies. The study was performed on 24 animals (12 control and 12 experimental). Gastric ulcers were induced by submucosal injections of 40% acetic acid solution into stomach submucosa and bilateral inferior vagal ganglia were collected one week afterwards. The number of galanin-immunoreactive perikarya in each ganglion was counted to determine fold-changes between both groups of animals and Q-PCR was applied to verify the changes in relative expression level of mRNA encoding both galanin and its receptor subtypes: GalR1, GalR2, GalR3. The results revealed a 2.72-fold increase in the number of galanin-immunoreactive perikarya compared with the controls. Q-PCR revealed that all studied genes were expressed in examined ganglia in both groups of animals. Statistical analysis revealed a 4.63-fold increase in galanin and a 1.45-fold increase in GalR3 mRNA as compared with the controls. No differences were observed between the groups for GalR1 or GalR2. The current study confirmed changes in the galaninergic inferior vagal ganglion response to stomach ulcerations and demonstrated, for the first time, the expression of mRNA encoding all galanin receptor subtypes in the porcine inferior vagal ganglia.

## Introduction

Galanin (Gal), a 29-amino-acid peptide firstly isolated from the porcine intestine [1], exerts its action on the peripheral tissues through three G protein-coupled transmembrane receptors: GalR1, GalR2 and GalR3. It is a neuropeptide of the central and peripheral nervous system

**Funding:** This publication was supported by RID (Project financially co-supported by Minister of Science and Higher Education in the range of the program entitled "Regional Initiative of Excellence" for the years 2019-2022). Project No. 010/RID/ 2018/19, amount of funding 12.000.000 PLN. The funders had no role in study design, data collection and analysis, decision to publish, or preparation of the manuscript.

**Competing interests:** No authors have competing interests.

involved in the neuronal regulation of the digestive system function under both physiological and pathological conditions. Studies have demonstrated that galanin is widely distributed throughout the gastrointestinal tract [2–6] and it is present both in the wall of gastrointestinal tract and the extrinsic nerves supplying the tract [7–10]. At the functional level, galanin plays a role in several biological processes in the digestive system [11–14] and, as demonstrated by the authors' previous studies, it modulates enteric nerve response to gastric ulcerations [15, 16].

The stomach is innervated by intrinsic (enteric) and extrinsic autonomic (parasympathetic, sympathetic) and sensory nerves, which compose a complex regulatory system. The enteric nervous system is characterized by high autonomy of action and is directly exposed to pathological processes ongoing in the stomach, while both intrinsic and extrinsic nerves play an important regulatory role in physiology and during the disease [17]. Sensory neurons are the first and particularly important link in an appropriate extrinsic neuronal regulation of the organ function. They are directly involved in receiving, transmitting and modulating peripheral information to the central nervous system. Neuropeptides (as substance P (SP), calcitonin gene-related peptide (CGRP), galanin and others) [18–22] synthetized by primary afferent neurons can be released from both central and peripheral nerve endings, affecting CNS and the periphery.

The stomach receives dual afferent innervation from sympathetic and parasympathetic nerves [23, 24], whose components differ in terms of function, perikaryon location and the extent of the innervated area. Primary afferent cell bodies of sympathetic nerves are located in the lower thoracic and upper lumbar dorsal root ganglia [7, 25–30] and reach the stomach via the sympathetic chain and the celiac plexus [24]. Parasympathetic primary afferent perikarya supplying stomach are located in bilateral nodose ganglia [7, 24, 30–32] and run within the vagus nerves.

Clinical studies have suggested that afferent fibers in the sympathetic nerves are involved in visceral pain signaling [24, 33] and the majority of galanin related experiments have been focused on such issues [34–40], while experiments on parasympathetic nodose afferents engaged in autonomic regulatory reflexes [24, 33] are very scarce [8, 9].

Gastric ulceration is a common stomach disorder affecting both humans and animals. In pigs, the presence of stomach ulcers has been reported since the 1960s [41, 42]. Gastric ulceration is accompanied by several other disorders, e.g. dyspepsia, delayed gastric emptying, maldigestion etc., suggesting impaired autonomic control of the stomach function, likely coregulated by galanin. The authors' previous studies revealed a complex intramural galaninergic response to pathological changes (gastric ulceration, colitis) in porcine stomach and intestines, pointing to the role of galanin in the enteric nervous system plasticity [15, 16, 43, 44]. Due to the crucial role of primary afferent vagal neurons in the extrinsic regulation of visceral reflexes [24], complementing the authors' previous studies, it was decided to examine changes in the number of galanin immunoreactive neurons in the nodose ganglia in the response to the disease (gastric ulcers). Despite the key role of receptors in the neuropeptide action, there is still no data on the distribution of galanin receptor subtypes in the porcine inferior vagal ganglia. For this reason, it was decided to examine the presence of Gal receptors by RT-PCR in the porcine inferior vagal ganglion. RT-PCR is the most reliable technique for this kind of studies, as reports have demonstrated that conventional immunohistochemical staining using antibodies against galanin receptors is not reliable and often produces false results [45].

Therefore, the aim of the present study was to evaluate the changes in the number of galanin immunoreactive neurons and to establish the changes in expression of mRNA encoding galanin and all of its receptor subtypes in the nodose ganglia of ulcered animals (in relation to controls).

It was decided to use pigs as the model since they are one of the best animal models for studies of human gastrointestinal tract diseases [46–48]. In addition, it is a husbandry animal of great economic value.

## Materials and methods

### Ethical regulations

The handling of animals and all experimental procedures were performed in accordance with the rules of the National Ethics Commission for Animal Experimentation (Polish Ministry of Science and Higher Education) and approved by the Local Ethics Committee of the University of Warmia and Mazury in Olsztyn (permission number 76/2012). For anesthesia/analgesia, animals were pre-treated with azaperone (Stresnil, Janssen Pharmaceutica, Belgium, 0.4 mg/kg b.w., i.m.) and atropine (Polfa, Poland, 0.04 mg/kg b. w., s.c.), and after 30 min they were anaesthetized with xylazine (Vetaxyl, Vet-Agro, Poland, 0.3 mg/kg b.w., i.m.) and ketamine (Bioketan, Vetoquinol Biowet, 15 mg/kg b.w., i.v., qs). At the final stage of the experiment, all the pigs were deeply anaesthetized (as described previously) and sacrificed with an overdose of anesthetic. All efforts were made to minimize animal suffering at each step of the experiment. All animal treatments were performed by a specialized veterinarian with appropriate knowledge and experience.

### Animals used in the study and experimental procedures

The experiment is a part of the wider study aimed at verifying the impact of gastric antral ulcers on complex nerve reactions. Thus, the study was performed on nodose ganglion tissues collected from animals used in the previous part of the experiment (sexually immature gilts of the Polish Large White breed, bodyweight approx. 20 kg, obtained from a commercial fattening farm, 14–260 Lubawa, Poland), focused on the intramural stomach neuron reaction to gastric ulcerations. The detailed descriptions of all experimental procedures are enclosed in these publications [15, 16, 49].

Briefly: 24 immature gilts (bodyweight ca. 20 kg) were assigned to experimental (n = 12) and control (n = 12) groups. In experimental animals, bilateral stomach ulcers were induced by submucosal injections of 1 cm$^3$ of 40% acetic acid solution into the anterior and posterior wall of the gastric antrum. After a 7-day period necessary to develop ulcers, both control and experimental animals were sacrificed. Half of the number of pigs in each cohort (control n = 6, experimental n = 6) designated as 'H-group', were transcardially perfused with 4% PFA (for immunohistochemistry). The rest of the animals formed 'M-group' (n = 6 control, n = 6 experimental) for molecular analysis (Q-PCR). Bilateral nodose ganglia were collected from both groups of animals and post-fixed according to the standard protocol (for H-group: immersed in 4% PFA for 30 min, rinsed in PBS and immersed in an 18% sucrose solution until they sank to the bottom of the container at 4˚C; for M-group: immersed in RNAlater for 24h at 4˚C and then frozen at -80 ˚C until further processing).

### Immunofluorescence

H-group tissues were cryo-sectioned along the long axis of the ganglion into 15-µm-thick consecutive slices and mounted on chrome alum-gelatin-coated numbered microscopic slides. A microscopic slide containing the central section [CS] of the ganglion tissue (determined from the total number of consecutive slides sectioned from the ganglion) and two additional slides with sections distanced about 300 µm laterally (LS) and medially (MS) from the CS (Fig 1) were subjected to the standard procedure of double immunostaining with a mixture of

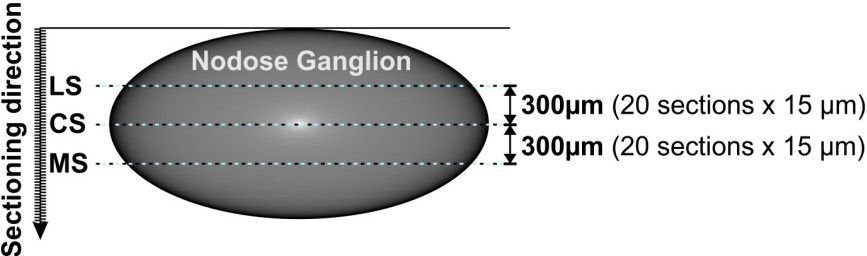

**Fig 1. Tissue sampling.** Drawing presenting the method of tissue sampling. Nodose ganglion tissue was cut (along its long axis) into consecutive 15μm sections and placed on numbered microscopic slides (four sections for each slide). The slide with a center section (CS) was determined from the total number of consecutive slides obtained from a ganglion by establishing its median. Two slides with sections distanced about 300 μm from central section (CS) were then determined by counting 20 consecutive sections (each section thickness = 15 μm; 4 sections on slide = 5 slides) in both directions (LS-lateral section; MS-medial section). This procedure guaranteed the collection of tissues from the same localizations in each ganglion (regardless of some individual differences in the size of the ganglia studied) and prevented the double counting of the same cells.

primary (rabbit anti-galanin, dilution 1:2500, code T4330, Peninsula Laboratories, San Carlos, CA, USA and mouse anti-PGP 9.5, dilution 1:600, code 7863–2004, clone 31A3, AbD Serotec, Eugene, OR, USA) and secondary (AlexaFluor 488, goat anti-mouse, dilution 1:500, code A11001 and AlexaFluor 555, goat anti-rabbit, dilution 1:500, code A-21428, Invitrogen, Kidlington, UK) antibodies. Both primary antibodies were recommended for application in the porcine tissues by suppliers and additional omission and replacement tests confirmed the specificity of the staining.

The number of Gal-immunoreactive perikarya in relation to all PGP-immunoreactive neurons was counted in three section planes of each ganglion. The axial center section and collateral sections, spaced apart by at least 300 μm (thus representing different depths of the ganglion structure) were analyzed. Such a procedure obtained representative results for the entire volume of each ganglion studied and enabled appropriate comparisons of the ganglions in all studied animals. This approach allowed a comprehensive verification of the reaction of the entire ganglion to gastric ulcerations and, simultaneously, to directly relate changes in peptide and gene expressions examined by immunocytochemistry and Q-PCR, respectively.

To obtain the most reliable results, confocal laser microscopy (Zeiss, LSM 700) microphotographs of the whole cross-sectional ganglion planes were taken using the tile scan function for both fluorescent channels. Additionally, the number of Gal-immunoreactive perikarya was verified in the entire cross-sectional area under a fluorescent microscope (Zeiss, Axio Scope. A1) equipped with a filter suitable for AlexaFluor555. Next, the images were analyzed using ImageJ software and the total number of PGP 9.5-immunoreactive and Gal-immunoreactive perikarya for each ganglion was counted. This procedure allowed for the analysis of at least 2,500 PGP-immunoreactive perikarya in each studied ganglion, providing at least 30,000 perikarya per each animal group. During the analysis, the investigator was blinded to the treated group—the tissue slides were delivered by a laboratory technician and unblinded to the investigator only after completing the evaluation. The obtained results were converted into percentages of Gal-immunoreactive perikarya present in the nodose ganglia of each animal. Finally, the results were presented as average percentages (± SEM) for the group of control and ulcered animals.

The dimensions of Gal-immunoreactive perikarya were measured using a confocal laser microscopy software analysis tool (Zeiss LSM Image Browser Ver. 4.2.0.121) on the group of 40 PGP 9.5/Gal-immunoreactive neurons in each animal group.

**Table 1. Sequences of primers used in the real-time PCR.**

| Gene | Sequences of primers | Sequence of origin (in Gene Bank) |
|---|---|---|
| GAPDH | Forward: GATCGTCAGCAATGCCTCCT | NM_001206359.1 |
| | Reverse: GATGCCGAAGTGGTCATGGA | |
| Gal | Forward: TGGGCCACATGCCATCGACA | NM_214234.1 |
| | Reverse: CGGCCTGGCTTCGTCTTCGG | |
| GalR1 | Forward: AGGATCACGGCGCACTGCCT | XM_003480426.2 |
| | Reverse: GGGATTCCTTGCCAATGTGGCACT | |
| GalR2 | Forward: GCCAAGCGCAAGGTAACGCG | XM_003484313.1 |
| | Reverse: GTAGGTGGCGCGGGTAAGCG | |
| GalR3 | Forward: GCACCACGCGCTCATCCTCT | XM_003355348.2 |
| | Reverse: AGACCAGCGGGTTGAGGCAG | |

Primers were designed using sequences of origin available in Gen Bank and Primer-BLAST software (http://ncbi.nlm.nih.gov).

## Real-time PCR

Nodose ganglia collected from the M-group animals were homogenized with fenozol and total RNA was isolated with a Total RNA Mini Plus kit (A&A Biotechnology, Poland) according to the manufacturer's manual. Reverse transcription was performed with 3.4 μg of total RNA and Maxima First Strand cDNA Synthesis Kit for RT-qPCR (code K1672, Thermo Fisher Scientific). Real-Time PCR was then performed from each cDNA sample with primers designed for porcine Gal, GalR1, GalR2, GalR3 and glyceraldehyde 3-phosphate dehydrogenase (GAPDH) genes. GAPDH has been tested and verified as an appropriate reference gene with stable expression level for both groups of animals. Sequences of primers (Table 1) were designed with Primer-BLAST software (http://ncbi.nlm.nih.gov) and sequence of origin available in GeneBank. PCR reactions were performed in triplets (for each cDNA sample) using the 7500 fast Real-Time PCR system (Applied Biosystems, USA) and SYBR® Select Master Mix (cat. No. 4472920, Applied Biosystems, USA) with the thermal profile consisting of: 2 min initial denaturation on 95 ˚C, 15 s denaturation on 95 ˚C, and 1 min annealing on 60 ˚C for 40 cycles. The data for Gal, GalR1, GalR2, GalR3 expression were normalized against GAPDH using software 7500 v. 2.0.2 (Applied Biosystems, USA). Relative expressions and $2^{-\Delta\Delta Cq}$ fold-change values were then calculated (primers with similar amplification efficiencies were used in the study).

## Statistical analysis

The results on mRNA fold-change, relative expression values and the number of immunofluorescent perikarya obtained from both animal groups were analyzed statistically with GraphPad Software Inc., USA, ver. 6 and appropriate tests (D'Agostino and Pearson omnibus normality test to verify Gaussian distribution; Student's t-test or Mann–Whitney U test for normal and non-normal distributed data, respectively; one-way ANOVA to compare differences between receptor subtype relative expressions) and considered to be significant at $P < 0.05$. The error bars represent a standard error of the mean (SEM).

## Results

### Analysis of double-immunolabeled sections

**Arrangement and characteristics of PGP 9.5/Gal-immunofluorescent vagal nodose perikarya in the control and experimental animals.** Microscopic analysis of

immunostained sections revealed that Gal-immunoreactive perikarya were scattered throughout the ganglia and no characteristic clusters were formed in the control (Fig 2A) or experimental (Fig 2B) animals. Most of the Gal-positive cell bodies were medium-to-highly fluorescent and measured about 37.21 ± 1.53 x 28.66 ± 1.30 μm in the control (Fig 2A-1 and 2A-3) and 37.50 ± 4.64 x 28.56 ± 1.26 in experimental pigs (Fig 2B-3 and 2B4, S1 Table), the size differences were not statistically significant between both animal groups. However, in both animal groups, occasional perikarya were larger (up to 65 x 60 μm) (Figs 2B-1 and 2B-2, 3A-1, 3A-2, 3A-3, 3C-1, 3C-2 and 3C-3) or smaller (Fig 3B-1, 3B-2, 3B3, 3D-1, 3D-2 and 3D-3) and featured characteristic clump-like pattern immunofluorescence (Fig 3A-2, 3C-2 and 3D-2).

**Changes in the number of PGP 9.5/Gal-immunofluorescent vagal nodose perikarya between control and experimental animals.** Gal-immunoreactive neuronal cell bodies accounted for 0.65 ± 0.11% of all PGP 9.5-positive ganglion perikarya examined in the control animals, while in the experimental pigs this percentage increased to 1.78 ± 0.32% (Figs 2A, 2B and 4; S2 Table). The resulting 2.72-fold increase in the number of Gal-immunoreactive perikarya in ulcer animals was statistically significant. Occasional Gal-immunoreactive nerve fibers were scattered irregularly in the nodose ganglia of the control (Fig 2A-2) and experimental (Fig 2B-3 and 2B-4) animals.

### Analysis of Q-PCR results

**Relative expression of studied genes (Gal, GalR1, GalR2, GalR3) in the nodose ganglia of the control and experimental animals.** Q-PCR results indicated the expression of mRNA encoding all studied genes in the vagal nodose ganglia of the control and experimental animals (Fig 5A, 5B, 5C; S3, S4, S5 and S6 Tables). The levels of mRNA expression for different receptor subtypes varied significantly within each group of animals. The relative expressions for GalR1 and GalR3 receptor subtypes were at comparable levels, while GalR2 was significantly less expressed (Fig 5B and 5C). Such a relationship appeared in both the control (Fig 5B) and experimental (Fig 5C) animals.

### Fold-change in relative expression of studied genes (Gal, GalR1, GalR2, GalR3) between control and experimental animals

Analysis of the Q-PCR results indicated a marked 4.63-fold elevation of Gal mRNA expression in the nodose ganglia of ulcered animals compared to controls (Fig 6A). In the group of galanin receptor subtypes, only GalR3 mRNA expression demonstrated a statistically significant 1.45-fold increase (Fig 6D), while changes in GalR1 (Fig 6B) and GalR2 (Fig 6C) receptor subtypes were not significant between experimental and control animals.

## Discussion

This study aimed to verify the galaninergic response of the inferior (nodose) vagal ganglion perikarya to gastric ulcerations. A significantly increased number of Gal-immunoreactive cells and elevated expression of genes encoding galanin and GalR3 receptor in the nodose ganglia of animals with stomach ulcerations were observed.

Vagal nerves are considered to play an important role in the regulation of stomach functions [50–54]. In the porcine stomach, vagal afferents constitute up to 80% of afferent pyloric innervation, while only a minority of cell bodies are dispersed within the bilateral spinal ganglia of the thoracic and lumbar neuromeres [7]. Studies on the stomach afferent innervation indicated that stomach pain is normally mediated by the afferent fibers in the sympathetic nerves, while vagal afferents mediate autonomic regulatory reflexes and special sensation, like

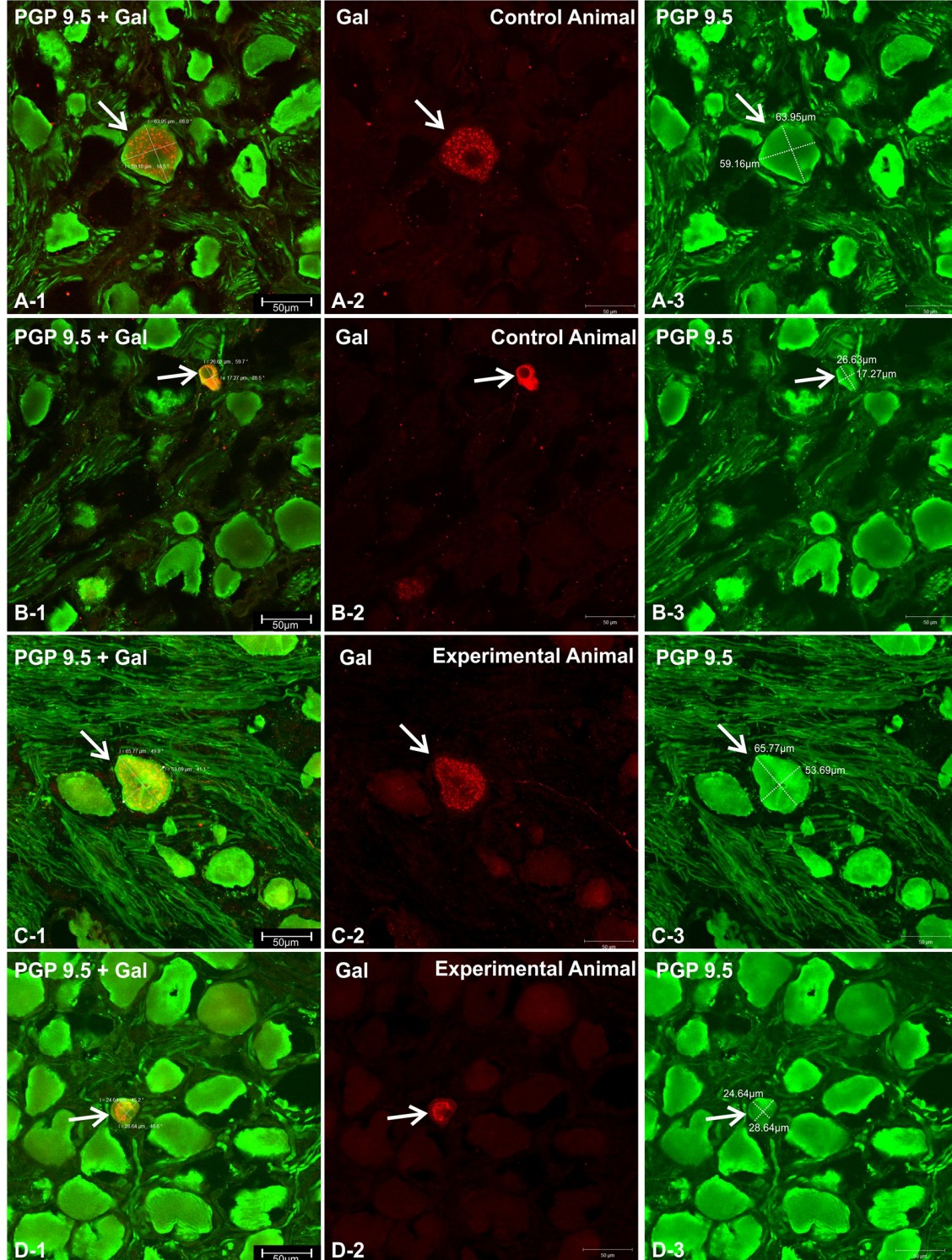

**Fig 2. Distribution and typical characteristics (shapes, immunofluorescence) of PGP 9.5/Gal-immunofluorescent perikarya within the longitudinal cross-sectional plane of the nodose ganglion in the control and experimental animals.** A set of microphotographs showing longitudinal cross-section of the representative inferior (nodose) vagal ganglion in the control (**A**) and experimental (**B**) animals, double immunostained with antibodies against PGP 9.5 (green channel) and galanin (red channel). The microphotographs present the overlap of both fluorescence channels. Figures (**A**) and (**B**) were created by combining a series of photomicrographs with the use of a confocal laser microscope tile scan function and present the entire cross-sectional surface of the

ganglion. High magnifications of selected areas (dotted line border from the picture (**A**) and (**B**)) are presented in the pictures below (**A-1, A-2, A-3, A-4**—control animals; **B-1, B-2, B-3, B-4**—experimental animals). PGP 9.5/Gal-immunoreactive perikarya are marked with arrows. Double arrows point to perikarya characterized by medium-to-weak Gal-immunofluorescence. Gal-immunofluorescent neuronal fibers are marked with arrowheads. Scale bars are included in the pictures.

satiety that follows food ingestion or fullness during gastric distension [24]. Therefore, some authors have even suggested classifying afferent neurons as visceral afferents "in sympathetic" or "in parasympathetic" nerves [24].

Recent studies, using ultramodern research techniques, have revealed that nodose neurons differ significantly in terms of gene expression pattern from other sensory neurons, i.e. DRG (dorsal root ganglion) and vagal jugular [55], indicating that their function is more complex than simple pain transduction. Moreover, during the developmental stages, nodose ganglion perikarya are derived from other tissues than spinal (DRG) and jugular ganglia cell bodies [56]. Therefore, changes in the number of galanin immunoreactive vagal nodose neurons and the simultaneous changes in the relative expressions of selected genes observed in the nodose ganglia of ulcered pigs are most likely related to autonomic regulation of the stomach function in pathological conditions and to the transduction of special kinds of sensation, like dyspepsia. Such neuronal reaction may form the basis for a complex pathophysiological body response to stomach ulcers.

Numerous studies have described the neurotransmitter expression changes (e.g. CGRP, nitric oxide synthase (NOS), tyrosine hydroxylase (TH), vasoactive intestinal peptide (VIP), neuropeptide Y (NPY)) in the nodose ganglia after experimental axotomy [57–61] and/or ligation/crush lesion of vagus nerve fibers [62], however only a few articles have described changes in galanin expression [63, 64]. Such experimental conditions interrupted axonal transportation and injured neuronal cells and directed synthetized neuropeptides towards trophic and protective functions. In contrast, since inflammatory processes affect the peripheral nerve terminals, neuronal cells are forced to orchestrate the body's response to inflammation. As a result, the data obtained in these different types of experiments (ligation/transection vs. inflammation) cannot be directly related.

Studies describing neuropeptide expression changes in the nodose perikarya during inflammation are rare and relate mainly to neurons supplying airways and lungs in bronchopulmonary inflammation, in which CGRP, SP and neurokinin A (NKA) expressions were changed [65, 66]. The changes in voltage sensitive currents observed in the nodose perikarya after multiple injections of acetic acid into the rat stomach wall clearly indicated the response of these neurons to gastritis [67–69], a phenomenon consistent with the authors' own observations. The expression of immune receptors for certain inflammatory chemokines, interleukins and interferons on vagal sensory neurons [70] reinforces the ability of these neurons to respond to inflammatory processes and coordinate pathophysiological reactions.

Although galanin expression changes in experimental models of inflammation were already observed in spinal sensory [34–38, 71] and trigeminal ganglia [39], the data on vagal nodose ganglia is very scarce. The administration of acetylsalicylic acid into the stomach resulted in a significant increase in the number of Gal-immunoreactive nodose perikarya [9], which is consistent with the authors' observations and confirms the significance of galanin in the neuroplasticity of the stomach supplying nodose neurons. In addition, such results clearly indicate the participation of galanin in the pathophysiology of gastritis, which is congruent with the inflammatory reactions observed in other parts of the digestive system [72].

Galanin acts as a modulator of inflammatory reactions, and its increasing production likely plays a role in limiting acute-phase inflammatory responses to prevent excessive inflammation

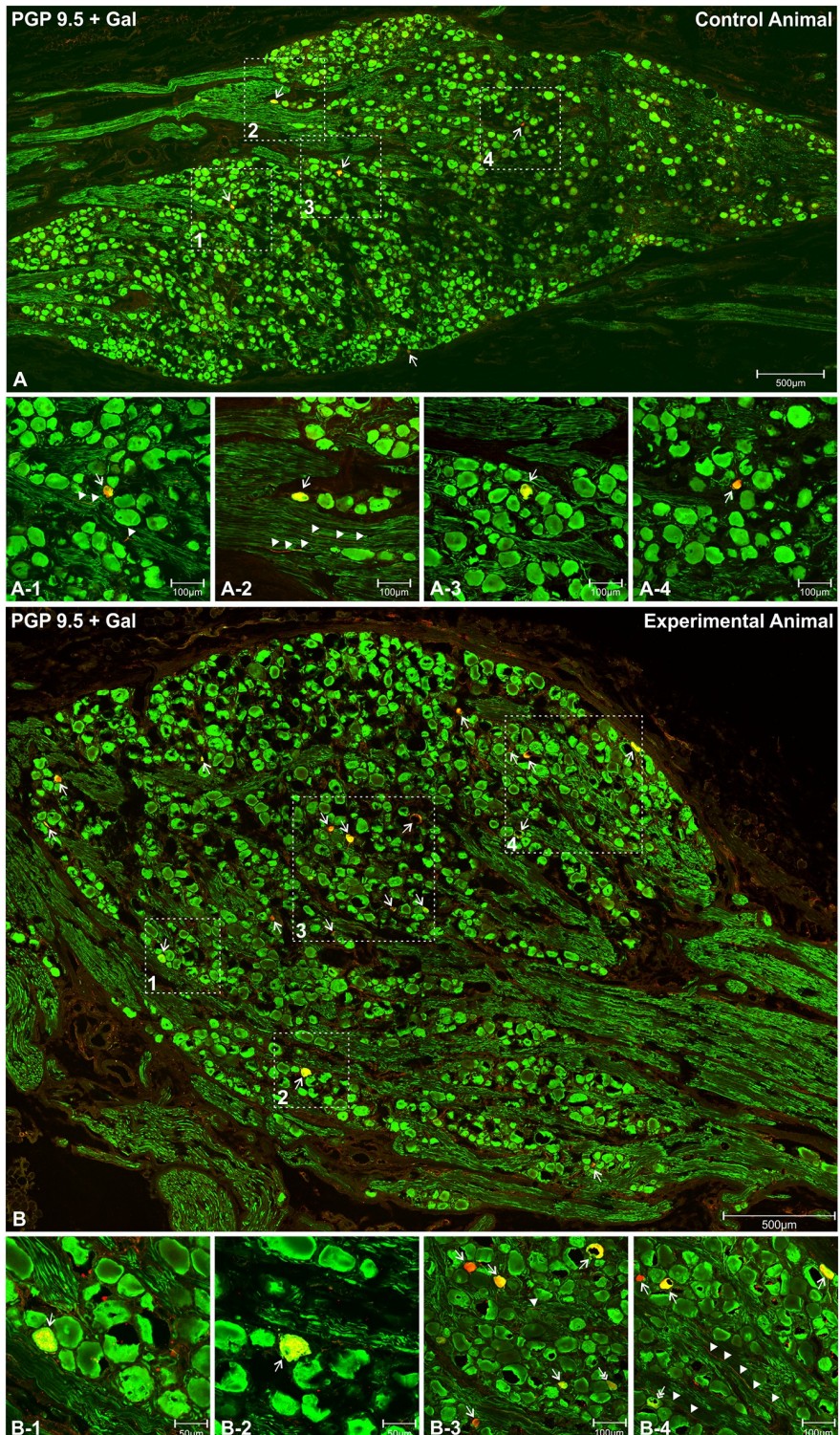

**Fig 3. Characteristics (shapes, immunofluorescence) of PGP9.5/Gal-immunofluorescent vagal nodose perikarya occasionally observed in the control and experimental animals.** Microphotographs showing high magnifications of occasionally observed Gal-immunoreactive neurons (arrows) with large (**A-1, A-3, A-3, C-1, C-2, C-3**) and small (**B-1, B-3, B-3, D-1, D-2, D-3**) cell body sizes occurring in the vagal nodose ganglia in the control (**A-1, A-3, A-3, B-1, B-3, B-3**) and experimental (**C-1, C-2, C-3, D-1, D-2, D-3**) animals. A characteristic, clump-like Gal-immunofluorescence pattern is visible in some of the perikarya (**A-1, A-3, A-3, C-1, C-2, C-3, D-1, D-2, D-3**), while others are characterized by strong, evenly dispersed immunofluorescence (**B-1, B-3, B-3**). Cell dimensions and scale bars are marked in the pictures.

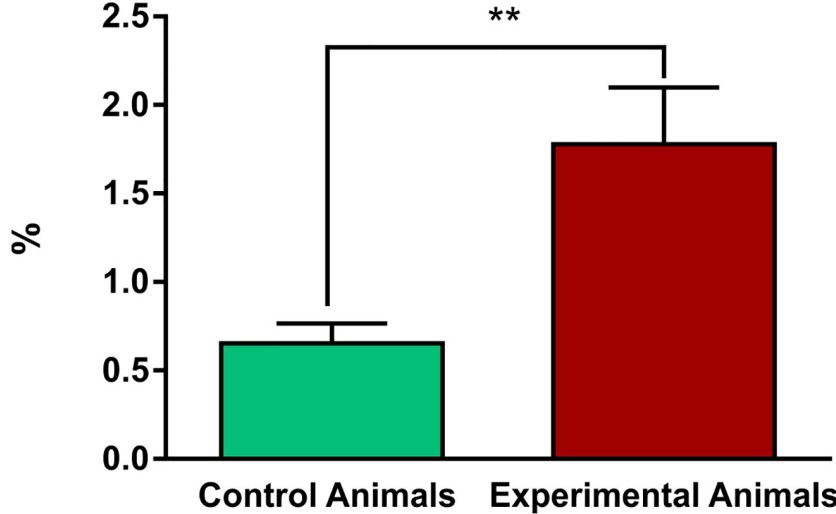

**Fig 4. Percentage differences in the nodose vagal ganglia PGP 9.5/Gal-immunofluorescent perikarya between control and experimental animals.** Graph showing the percentage of PGP/Gal-immunofluorescent perikarya observed in the nodose vagal ganglia in the control and experimental animals. The increase was statistically significant (** p<0.005; error bars indicate SEM).

and restore homeostasis [72]. Such galanin action is mediated by the reduction of tissue sensitivity to neurogenic inflammatory factors (as SP, CGRP) or by the regulation of proinflammatory cytokine production [72]. The acute gastric ulcerations examined in the current study were accompanied by a strong inflammatory response, thus it might be speculated that galaninergic nerve response may be additionally associated with the regulation of the inflammatory process itself. Therefore, the increased number of Gal-immunofluorescent perikarya supplying the ulcered stomach, observed in both the gastric wall [15, 16] and nodose ganglia (present article [9]), suggests such an inflammatory modulating action of galanin in ulcer disease. Since GalR3 mediates anti-edema effects in the skin [73], its increased expression observed in the current study ameliorates the thesis on the galanin-mediated inhibition of plasma extravasation in ulcered tissue to reduce neurogenic inflammation. The hypothesis of the anti-inflammatory function of galanin in acute ulcer disease is strongly supported by experiments demonstrating its protective and anti-inflammatory effects in an acute inflammatory phase of experimentally induced colitis [74, 75].

A simultaneous peptide and gene increase (observed in the current study) indicates an upregulation of galanin synthesis by the nodose vagal perikarya and suggests its transportation to the stomach and CNS (central nervous system) via descending and ascending nerve branches, respectively.

By comparing the small absolute number of Gal-immunoreactive vagal nodose neurons to a much larger number of intramural galaninergic perikarya observed in the porcine stomach [15, 16] it appears that most of these nodose perikarya are mainly responsible for galanin release toward the central nervous system. The first CNS site for synaptic contact of the primary vagal afferent neurons is the nucleus tractus solitarii (NTS), which integrates the sensory inputs from multiple brain regions to arrange a complex nerve regulatory network of innervated structures under physiological and pathological conditions. The authors' previous results

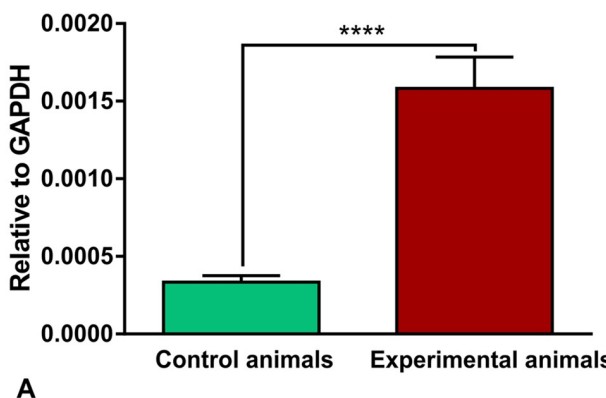

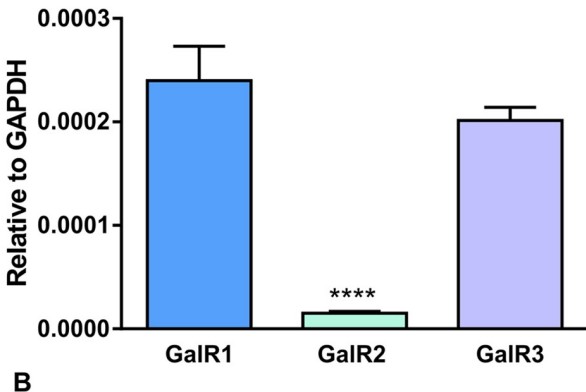

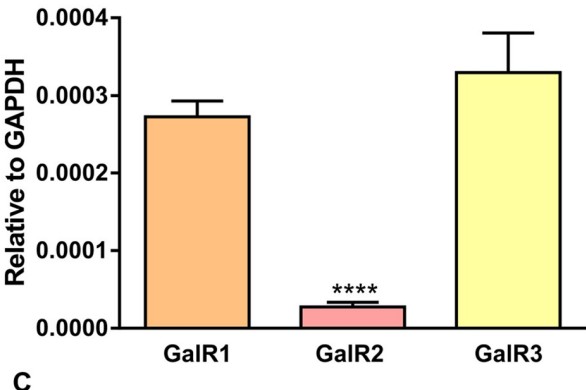

**Fig 5. Relative expression of mRNA encoding Gal, GalR1, GalR2, GalR3 in the nodose ganglia of the control and experimental animals.** Set of graphs showing the relative expression (to GAPDH as a housekeeping gene) of mRNA encoding galanin (**A**) and galanin (GalR1, GalR2, GalR3) receptors (**B, C**) in the control and experimental animals. The expression of galanin was markedly increased in experimental animals in relation to controls (**A**). GalR2 was significantly less expressed in relation to GalR1 and GalR3 in the group of control (**B**) and experimental (**C**) animals. (**** $p < 0.0001$; error bars indicate SEM).

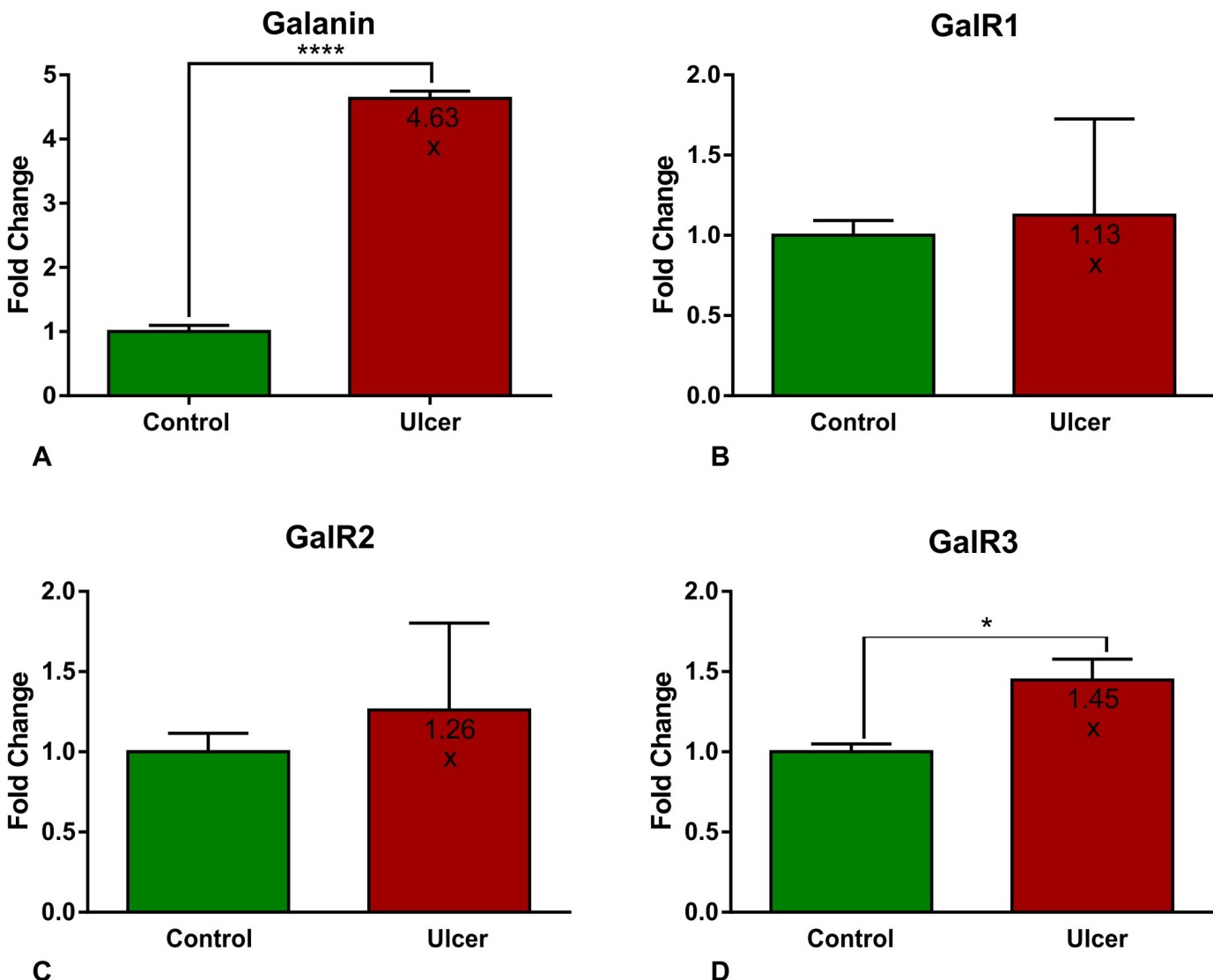

**Fig 6. Fold-change in mRNA expression of Gal, GalR1, GalR2, GalR3 encoding genes between control and experimental animal nodose ganglia.** Set of graphs showing fold-change in mRNA expression of genes encoding galanin (**A**) and GalR1 (**B**), GalR2 (**C**), GalR3 (**D**) receptors between control and experimental animals. The expression of galanin (**A**) and GalR3 (**D**) was significantly elevated in the nodose vagal ganglia of experimental animals in relation to controls, while changes in GalR1 (**B**) and GalR2 (**C**) were not statistically significant (* $p < 0.05$; **** $p < 0.0001$; error bars indicate SEM).

have indicated pyloric projecting perikarya in the porcine bilateral DMX [31] and in the vagal nodose ganglia [7, 9]. Importantly, galanin and GalR1 receptor were found in NTS [76, 77], indicating the potential role of galaninergic interaction between central nerve terminals of the vagal nodose neurons and NTS perikarya. Microinjections of galanin into NTS markedly reduced cough reflexes [78], which, due to the similarities between characteristics of central processing of nociceptive and cough-related inputs, suggests the inhibitory function of galanin released from central nerve endings of the nodose neurons. The view that GalR1 activation mainly mediates an analgesic effect [79–82] seems to be in line with such an assumption. However, reports indicating that the vagus nerve is engaged in the conduction of a special kind of sensation like dyspepsia [24], suggest that, in ulcered animals, galanin released centrally from the nodose perikarya acts as a defense mechanism against the over-consumption of food. A

significant reduction in food intake after GALP (galanin like peptide) injection into the NTS [83] strongly supports the presented hypothesis and such orexigenic function may protect the ulcered mucosa. Nevertheless, further studies are needed to elucidate the exact role of galanin released into the NTS on the stomach function.

The present study has shown, for the first time, the expression of mRNA encoding all types of galanin receptors in the porcine nodose ganglia, whose perikarya transport receptor proteins to the periphery. Although the presence of GalR1 and GalR2 receptors in sensory neurons was described by various researchers [84–86] and data on GalR3 was also discussed [87, 88], only a few articles describe their expression in vagal nodose neurons [89]. Galanin reduces neuronal excitability via GALR1 and GALR3 receptor subtypes and increases excitability via subtype GALR2. The divergent, excitatory, and inhibitory effects of galanin observed in gastro-esophageal vagal reflexes [90] suggest variable expression of galanin receptors subtypes on sensory neurons. In the authors' studies, expressions of all receptor subtypes (GalR1-3) were found in the nodose ganglia, which is convergent with data obtained in mice [91]. Importantly, GalR1 and GalR3 were expressed at similar levels, while GalR2 was at a much lower level as compared to other receptors, and such a tendency was observed in both healthy and ulcered animals. Importantly, these results differ significantly from the data presented in mice [91], indicating interspecies differences between rodents and omnivores. The very low level of GalR2 observed in both groups of studied pigs suggests its minor importance for the porcine nodose neuron function in physiology and disease. Such an assumption seems to be in line with the speculation presented by Page [91] that agonists of GalR1 or GalR3 may have more therapeutic potential for reducing gastric mechanosensitivity than GalR2 antagonists.

Tissue changes induced by gastric ulcerations directly influenced the vagal peripheral afferent endings. These endings, as demonstrated in previous studies, can be categorized into two main functional groups: mucosal receptors responding to light stroking of the mucosa and chemical stimuli, and mechanoreceptors responding to the tension of gastric wall [92, 93]. The majority of mechanoreceptors belong to Intraganglionic Laminar Endings (IGLEs) (located within the myenteric ganglia) [94] and intramuscular arrays (IMAs) [95–97].

It seems to be reasonable to speculate that the increased number of myenteric Gal-positive neurons observed in different stomach localizations under antral ulcerations [15, 16] influence the nodose neurons peripheral IGLEs and strongly stimulate their GalR1 receptor.

The current experiment revealed that GalR3 was the only receptor whose expression was changed (increased) in ulcered animals, indicating its involvement in the nodose ganglia response to stomach ulcer. Interestingly, GalR3 had no functional involvement in the regulation of tension and mucosal gastric vagal receptor mechanosensitivity [91], thus, it is probably involved in other kinds of afferent sensation (as previously suggested dyspepsia). Moreover, it cannot be excluded that galanin released in the NTS during an inflammatory reaction participates in the CNS-originating downregulation of vagal nodose neurons activity.

Due to the GalR1 and GalR3 inhibitory signaling pathways [98], the results obtained in the present study suggest a mainly inhibitory function of galanin-stimulated nodose perikarya. Similarly, an experiment in mice demonstrated that exogenous galanin predominantly developed inhibitory effects on neuronal reflexes [90]. However, future studies are needed to verify the presented assumptions or to find another explanation for the role of GalR3 in vagal nodose neurons.

## Conclusions

The current study has demonstrated, for the first time, a galaninergic response of the vagal nodose perikarya to gastric ulcerations. This study has also confirmed the role of galanin and

GalR3 receptor in the neuronal plasticity of primary afferent vagal perikarya—neurons which are known to be responsible for controlling extrinsic autonomic reflexes regulating the function of internal organs. Moreover, the expression of mRNA encoding all galanin receptor subtypes in the porcine vagal nodose ganglia was demonstrated as well as differences in their expression levels.

The current results complement the authors' previous research [15, 16] and lay the groundwork for future studies on the role of galanin in gastric ulcer disease.

## Supporting information

**S1 Table. Data set—Cell dimensions.**
(PDF)

**S2 Table. Data set—Number and percentage of PGP 9.5 + / Gal + immunoreactive neurons.**
(PDF)

**S3 Table. Data set—Real-time Ct—Values for mRNA encoding Gal.**
(PDF)

**S4 Table. Data set—Real-time Ct—Values for mRNA encoding GalR1 receptor.**
(PDF)

**S5 Table. Data set—Real-time Ct—Values for mRNA encoding GalR2 receptor.**
(PDF)

**S6 Table. Data set—Real-time Ct—Values for mRNA encoding GalR3 receptor.**
(PDF)

## Acknowledgments

The authors would like to thank Ms. Adrianna Plywacz for technical assistance in laboratory work.

## Author Contributions

**Conceptualization:** Michal Zalecki.

**Data curation:** Michal Zalecki.

**Formal analysis:** Marzena Mogielnicka-Brzozowska.

**Funding acquisition:** Jerzy Kaleczyc.

**Investigation:** Michal Zalecki, Zenon Pidsudko, Marzena Mogielnicka-Brzozowska, Amelia Franke-Radowiecka.

**Methodology:** Michal Zalecki, Judyta Juranek, Zenon Pidsudko, Amelia Franke-Radowiecka.

**Project administration:** Michal Zalecki.

**Resources:** Jerzy Kaleczyc.

**Software:** Michal Zalecki.

**Supervision:** Marzena Mogielnicka-Brzozowska, Amelia Franke-Radowiecka.

**Validation:** Judyta Juranek.

**Visualization:** Michal Zalecki.

**Writing – original draft:** Michal Zalecki.

**Writing – review & editing:** Judyta Juranek, Jerzy Kaleczyc.

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
