## [Decision Letter · Decision Letter 0]

1 Sep 2020

PONE-D-20-20776

Inferior vagal ganglion galaninergic response to gastric ulcers

PLOS ONE

Dear Dr. Zalecki,

Thank you for submitting your manuscript to PLOS ONE. After careful consideration, we feel that it has merit but does not fully meet PLOS ONE’s publication criteria as it currently stands. Therefore, we invite you to submit a revised version of the manuscript that addresses the points raised during the review process.

Please address the reviewer's concerns, both regarding manuscript editing and methodologies. The topography of your staining is of particular concern, as the various territories are differentially represented. 

We look forward to receiving your revised manuscript.

Kind regards,

Tudor C Badea, M.D., M.A., Ph.D.

Academic Editor

PLOS ONE

Journal Requirements:

2. To comply with PLOS ONE submissions requirements, in your Methods section, please provide additional information on the animal research and ensure you have included details on (1) methods of sacrifice, (2) methods of anesthesia and/or analgesia, and (3) efforts to alleviate suffering

Additional Editor Comments (if provided):

Please address the reviewer's concerns, both regarding manuscript editing and methodologies. The topography of your staining is of particular concern, as the various territories are differentially represented.

Reviewers' comments:

Reviewer's Responses to Questions

**Comments to the Author**

1. Is the manuscript technically sound, and do the data support the conclusions?

Reviewer #1: Partly

2. Has the statistical analysis been performed appropriately and rigorously? 

Reviewer #1: Yes

3. Have the authors made all data underlying the findings in their manuscript fully available?

Reviewer #1: Yes

4. Is the manuscript presented in an intelligible fashion and written in standard English?

Reviewer #1: Yes

5. Review Comments to the Author

Reviewer #1: In the present submission, Zalecki and colleagues examined changes in the distribution of galanin in the inferior vagal ganglion in a piglet model of gastric ulcer. The authors report that, one week following submucosal injection of acetic acid solution, there was a significant increase in galanin-IR perikarya and a significant increase in in GalR3 mRNA compared to untreated animals.

A series of issues must be addressed:

An ASA-induced increase in galanin-IR in the nodose ganglion was already reported (PMID: 26917098).

A major portion of the Introduction is very descriptive and naïve (lines 62-70; 75-81; 89-93; 97-107) and could be easily condensed, likewise for the Discussion.

What criterion was used to determine which of the ganglion slices was utilized for counting? If the slices were cut in a consecutive manner, which criterion was used to avoid double counting? This appears to be a relevant issue since the soma dimensions varied significantly (17-66 and 11-50 in the X and Y axis, respectively).

Within the nodose ganglion there is a loose viscerotopic representation of the upper GI tract (PMID: 2738198), however, the localization of galanin does not appear to be localized in particular areas. The authors should consider retrograde labelling from the stomach + IHC for galanin.

Please specify whether the perikarya were counted by an investigator unaware of the treatment.

Galanin neurotransmission has been associated with motility changes in the upper GI tract, however, its relation with gastric ulcers is unclear and should be discussed.

The manuscript should be edited for grammar.

6. PLOS authors have the option to publish the peer review history of their article (what does this mean?). If published, this will include your full peer review and any attached files.

Reviewer #1: **Yes: **R.Alberto Travagli

---

## [Author Response · Author response to Decision Letter 0]

28 Sep 2020

The authors would like to kindly thank the Reviewer for evaluating the manuscript and for such precise review and all the critical comments. All efforts were made to fulfill all the Reviewer’s expectations. We have referred to all the requirements raised by the Reviewer. We strongly believe that all the amendments performed have enriched the quality of paper and in the present form it would be accepted for publication in PlosOne. 

Here is the list of answers and corrections required, for all the comments made by the Reviewer:

1. An ASA-induced increase in galanin-IR in the nodose ganglion was already reported (PMID: 26917098).

We would like to kindly inform that the article was shortly mentioned in the previous version of the manuscript (citation no 9). However, in the improved version of our manuscript we have related our results to those presented by authors in the cited article (Discussion).

2. A major portion of the Introduction is very descriptive and naïve (lines 62-70; 75-81; 89-93; 97-107) and could be easily condensed, likewise for the Discussion.

According to the Reviewer’s suggestion we have modified and condensed indicated paragraphs of the Introduction, and some paragraphs in the Discussion. All changes are outlined <highlighted> in the manuscript with tracked changes (blue color).

3. What criterion was used to determine which of the ganglion slices was utilized for counting? If the slices were cut in a consecutive manner, which criterion was used to avoid double counting? This appears to be a relevant issue since the soma dimensions varied significantly (17-66 and 11-50 in the X and Y axis, respectively).

The authors agree that the original description was not fully legible, and it was not clear which sections were examined. We would like to kindly explain, that in our experiment we decided to apply methodology which will be most uniformly comparable between all animals studied to obtain most relevant results. Therefore, the nodose ganglion was cut along its long axis into 15 µm consecutive sections and mounted on microscopic, numbered slides (4 sections for each microscopic slide). Then, we established the median slide (from the total number of consecutive slides obtained from the ganglion) - and determined the “central section” in it. Next we determined two slides with sections distanced about 300 µm from “central section” by counting 20 consecutive sections (each section thickness =15 µm; 4 sections on slide = 5 slides) in both directions (LS-lateral section; MS-medial section) from central section. Finally, we obtained 3 representative sections from each ganglion studied which were distanced at least 300 µm from each other. Such procedure guaranteed the examination of tissues from the same localizations in each ganglion (regardless of some individual differences in the size of the ganglia) and prevented the double counting of the same cells.

In order to clearly explain the issue, an additional text (in the manuscript) and a new drawing (Fig 1) with a corresponding legend were included into the manuscript. We would like to add, that the inclusion of an additional figure has changed the numbering of the remaining figures (as indicated in the manuscript with tracked changes).

4. Within the nodose ganglion there is a loose viscerotopic representation of the upper GI tract (PMID: 2738198), however, the localization of galanin does not appear to be localized in particular areas. The authors should consider retrograde labelling from the stomach + IHC for galanin.

We would like to kindly explain that vagal nodose perikarya supplying stomach are mostly unevenly scattered within the ganglion, and no characteristic clusters are formed. The authors of the quoted article (PMID: 2738198) observed “crude” viscerotopic organization of perikarya supplying soft palate, pharynx, and esophagus (which was supplied from jugular and nodose ganglia) while stomach projecting perikarya were diffusely scattered throughout the nodose ganglion area (Fig 4B page 254). Importantly, all stomach projecting cells were observed exclusively within the nodose ganglion – what is consistent with the area of our research. We would like to add, that our previous tracing studies revealed that nodose neurons supplying the pyloric sphincter in the pig were also unevenly scattered throughout the ganglion (PMID: 23103024).

Since, it is not technically possible to inject the tracer into the entire stomach wall (due to the dimensions of the porcine stomach), the application of neuronal tracing method to visualize gastric nodose perikarya in the present experiment would have limited further analysis exclusively to neurons supplying the localizations of tracer depositions. It should be highlighted that gastric ulcerations are associated with complex stomach function disorders (e.g. dyspepsia, delayed gastric emptying, maldigestion) indicating dysregulation of extrinsic autonomic reflexes covering the wide area of the stomach wall. Our previous results confirmed a wide response of the stomach wall neurons to antral ulcerations (PMID: 27175780, PMID: 29717796; PMID: 30155561) and demonstrated the influence of ulcerations on the number of traced neurons (PMID: 25962176). 

Moreover, the time for tracer retrograde transportation to the far distanced nodose vagal perikarya in the pig requires at least 14-21 days to obtain properly saturated perikarya – and such period significantly exceeds the time of ulcer development. 

Therefore, we decided to verify galaninergic changes in the uniformly selected nodose ganglion tissue slides, which seems to be highly advisable method in this study. Our approach allowed to comprehensively verify the reaction of the entire ganglion to gastric ulcerations and, simultaneously, to directly relate changes in peptide and gene expressions examined by immunocytochemistry and Q-PCR, respectively.

We strongly believe that Reviewer accepts our rationale for taking such an approach. 

5. Please specify whether the perikarya were counted by an investigator unaware of the treatment.

We would like to kindly inform that during the analysis the investigator was unaware of the treatment.

Appropriate information has been incorporated into the Materials and Methods:

During the analysis, the investigator was blinded to the treated group — tissue slides were delivered by laboratory technician and unblinded to the investigator only after completing the evaluation.

6. Galanin neurotransmission has been associated with motility changes in the upper GI tract, however, its relation with gastric ulcers is unclear and should be discussed.

We would like to kindly explain that so far the role of galanin in the nodose ganglia is poorly understood and there are no literature data on its function during stomach pathology. As we discussed earlier, nodose ganglia are mainly responsible for regulation of autonomic reflexes, while dorsal root ganglia and vagal jugular ganglia are pain transduction, thus pain modulating function of galanin, proposed in some of the inflammatory experiments seems to be of minor importance. Therefore, as we proposed in our paper, galanin can be engaged in the transduction of special kind of sensation like dyspepsia, as a defense mechanism against consuming too much food in ulcer disease. 

However, as the Reviewer correctly noted, according to many studies galanin acts as a modulator of inflammatory reaction, and its increasing production is considered to be aimed at limiting acute-phase inflammatory responses to prevent excessive inflammation and to restore homeostasis (PMID: 21087790). Such galanin action is mediated by reduction of tissue sensitivity to neurogenic inflammation factors (as SP, CGRP) or regulation of proinflammatory cytokine production (PMID: 21087790). Gastric ulcer model examined in our study was accompanied by a strong inflammatory response, thus galaninergic nerve response may be additionally associated with the regulation of the inflammatory process itself. Therefore, increased number of Galanin-immunofluorescent perikarya supplying the ulcered stomach, observed in both, the gastric wall (PMID: 29717796, PMID: 27175780 ) and nodose ganglia (present article, PMID: 26917098), suggests such inflammation modulating action of galanin in ulcer disease. Since GalR3 mediates antiedema effect in the skin (PMID: 18679831), its increased expression observed in our study ameliorates the thesis on the galanin mediated inhibition of plasma extravasation in ulcered tissue and inhibiting neurogenic inflammation. The hypothesis on anti-inflammatory function of galanin in acute ulcer disease is strongly supported by experiments demonstrating its protective and anti-inflammatory effects in acute inflammatory phase of experimentally induced colitis (PMID: 16846834, PMID: 17331599). 

Such paragraph has been incorporated into the Discussion.

7. The manuscript should be edited for grammar. 

The manuscript has been verified by a native speaker.

In addition, we would like to kindly inform, that after consultation and approval by the PLOS ONE Academic Editor, we have incorporated as a co-author the name of Professor Jerzy Kaleczyc, the Head of the Department of Animal Anatomy, who managed all the team, helped us to thoroughly revise the article and to raise funds for the publication fee. His name was already mentioned in the Acknowledgments in the original version of the manuscript.

---

## [Decision Letter · Decision Letter 1]

16 Oct 2020

PONE-D-20-20776R1

Inferior vagal ganglion galaninergic response to gastric ulcers

PLOS ONE

Dear Dr. Zalecki,

Thank you for submitting your manuscript to PLOS ONE. After careful consideration, we feel that it has merit but does not fully meet PLOS ONE’s publication criteria as it currently stands. Therefore, we invite you to submit a revised version of the manuscript that addresses the points raised during the review process.

Although the reviewer handed a "reject" verdict, the expressed critique, consisting of grammar check and better pathophysiology explanation (ideally in the discussion), really warrant a minor revision. 

Please strive to address these issues, by working with a scientific english expert, and providing the desired context. 

We look forward to receiving your revised manuscript.

Kind regards,

Tudor C Badea, M.D., M.A., Ph.D.

Academic Editor

PLOS ONE

Additional Editor Comments (if provided):

Although the reviewer handed a "reject" verdict, the expressed critique, consisting of grammar check and better pathophysiology explanation (ideally in the discussion), really warrant a minor revision.

Please strive to address these issues, by working with a scientific english expert, and providing the desired context.

Reviewers' comments:

Reviewer's Responses to Questions

**Comments to the Author**

1. If the authors have adequately addressed your comments raised in a previous round of review and you feel that this manuscript is now acceptable for publication, you may indicate that here to bypass the “Comments to the Author” section, enter your conflict of interest statement in the “Confidential to Editor” section, and submit your "Accept" recommendation.

Reviewer #1: (No Response)

2. Is the manuscript technically sound, and do the data support the conclusions?

Reviewer #1: Yes

3. Has the statistical analysis been performed appropriately and rigorously? 

Reviewer #1: Yes

4. Have the authors made all data underlying the findings in their manuscript fully available?

Reviewer #1: Yes

5. Is the manuscript presented in an intelligible fashion and written in standard English?

Reviewer #1: No

6. Review Comments to the Author

Reviewer #1: The manuscript has improved, however, the grammar still needs to be improved and the content of the manuscript should be better placed in a pathophysiological contest.

7. PLOS authors have the option to publish the peer review history of their article (what does this mean?). If published, this will include your full peer review and any attached files.

Reviewer #1: No

---

## [Author Response · Author response to Decision Letter 1]

5 Nov 2020

The authors would like to kindly thank the Editor for decision on giving us a chance to deal with the second revision. 

Both suggestions were included into the revised version of the article.

1. In order to improve the quality of the article in the best possible way, the manuscript was checked in two stages: firstly, it was checked by an English-speaking scientist, then, it was submitted to professional translation office dealing with the correction of scientific texts. We attach the certificate obtained from the translation office. All corrections are depicted (by colors) in the “Track Changes” manuscript file.

2. In order to better emphasize the pathophysiological context, the discussion was modified, re-arranged and supplemented with additional sentences (e.g. lines 344-345, 363-366, 373-392 – manuscript with track changes). We would like to add, that the inclusion of such modifications has changed the numbering of article citations (as indicated in the manuscript with track changes).

 We believe that all the corrections incorporated have enriched the quality of the paper and in the present form it would be accepted for publication in your valuable journal.

We would kindly appreciate the acceptance of this manuscript for publication

Review 1 (28.09.2020)

The authors would like to kindly thank the Reviewer for evaluating the manuscript and for such precise review and all the critical comments. All efforts were made to fulfill all the Reviewer’s expectations. We have referred to all the requirements raised by the Reviewer. We strongly believe that all the amendments performed have enriched the quality of paper and in the present form it would be accepted for publication in PlosOne. 

Here is the list of answers and corrections required, for all the comments made by the Reviewer:

1. An ASA-induced increase in galanin-IR in the nodose ganglion was already reported (PMID: 26917098).

We would like to kindly inform that the article was shortly mentioned in the previous version of the manuscript (citation no 9). However, in the improved version of our manuscript we have related our results to those presented by authors in the cited article (Discussion).

2. A major portion of the Introduction is very descriptive and naïve (lines 62-70; 75-81; 89-93; 97-107) and could be easily condensed, likewise for the Discussion.

According to the Reviewer’s suggestion we have modified and condensed indicated paragraphs of the Introduction, and some paragraphs in the Discussion. All changes are outlined <highlighted> in the manuscript with tracked changes (blue color).

3. What criterion was used to determine which of the ganglion slices was utilized for counting? If the slices were cut in a consecutive manner, which criterion was used to avoid double counting? This appears to be a relevant issue since the soma dimensions varied significantly (17-66 and 11-50 in the X and Y axis, respectively).

The authors agree that the original description was not fully legible, and it was not clear which sections were examined. We would like to kindly explain, that in our experiment we decided to apply methodology which will be most uniformly comparable between all animals studied to obtain most relevant results. Therefore, the nodose ganglion was cut along its long axis into 15 µm consecutive sections and mounted on microscopic, numbered slides (4 sections for each microscopic slide). Then, we established the median slide (from the total number of consecutive slides obtained from the ganglion) - and determined the “central section” in it. Next we determined two slides with sections distanced about 300 µm from “central section” by counting 20 consecutive sections (each section thickness =15 µm; 4 sections on slide = 5 slides) in both directions (LS-lateral section; MS-medial section) from central section. Finally, we obtained 3 representative sections from each ganglion studied which were distanced at least 300 µm from each other. Such procedure guaranteed the examination of tissues from the same localizations in each ganglion (regardless of some individual differences in the size of the ganglia) and prevented the double counting of the same cells.

In order to clearly explain the issue, an additional text (in the manuscript) and a new drawing (Fig 1) with a corresponding legend were included into the manuscript. We would like to add, that the inclusion of an additional figure has changed the numbering of the remaining figures (as indicated in the manuscript with tracked changes).

4. Within the nodose ganglion there is a loose viscerotopic representation of the upper GI tract (PMID: 2738198), however, the localization of galanin does not appear to be localized in particular areas. The authors should consider retrograde labelling from the stomach + IHC for galanin.

We would like to kindly explain that vagal nodose perikarya supplying stomach are mostly unevenly scattered within the ganglion, and no characteristic clusters are formed. The authors of the quoted article (PMID: 2738198) observed “crude” viscerotopic organization of perikarya supplying soft palate, pharynx, and esophagus (which was supplied from jugular and nodose ganglia) while stomach projecting perikarya were diffusely scattered throughout the nodose ganglion area (Fig 4B page 254). Importantly, all stomach projecting cells were observed exclusively within the nodose ganglion – what is consistent with the area of our research. We would like to add, that our previous tracing studies revealed that nodose neurons supplying the pyloric sphincter in the pig were also unevenly scattered throughout the ganglion (PMID: 23103024).

Since, it is not technically possible to inject the tracer into the entire stomach wall (due to the dimensions of the porcine stomach), the application of neuronal tracing method to visualize gastric nodose perikarya in the present experiment would have limited further analysis exclusively to neurons supplying the localizations of tracer depositions. It should be highlighted that gastric ulcerations are associated with complex stomach function disorders (e.g. dyspepsia, delayed gastric emptying, maldigestion) indicating dysregulation of extrinsic autonomic reflexes covering the wide area of the stomach wall. Our previous results confirmed a wide response of the stomach wall neurons to antral ulcerations (PMID: 27175780, PMID: 29717796; PMID: 30155561) and demonstrated the influence of ulcerations on the number of traced neurons (PMID: 25962176). 

Moreover, the time for tracer retrograde transportation to the far distanced nodose vagal perikarya in the pig requires at least 14-21 days to obtain properly saturated perikarya – and such period significantly exceeds the time of ulcer development. 

Therefore, we decided to verify galaninergic changes in the uniformly selected nodose ganglion tissue slides, which seems to be highly advisable method in this study. Our approach allowed to comprehensively verify the reaction of the entire ganglion to gastric ulcerations and, simultaneously, to directly relate changes in peptide and gene expressions examined by immunocytochemistry and Q-PCR, respectively.

We strongly believe that Reviewer accepts our rationale for taking such an approach. 

5. Please specify whether the perikarya were counted by an investigator unaware of the treatment.

We would like to kindly inform that during the analysis the investigator was unaware of the treatment.

Appropriate information has been incorporated into the Materials and Methods:

During the analysis, the investigator was blinded to the treated group — tissue slides were delivered by laboratory technician and unblinded to the investigator only after completing the evaluation.

6. Galanin neurotransmission has been associated with motility changes in the upper GI tract, however, its relation with gastric ulcers is unclear and should be discussed.

We would like to kindly explain that so far the role of galanin in the nodose ganglia is poorly understood and there are no literature data on its function during stomach pathology. As we discussed earlier, nodose ganglia are mainly responsible for regulation of autonomic reflexes, while dorsal root ganglia and vagal jugular ganglia are pain transduction, thus pain modulating function of galanin, proposed in some of the inflammatory experiments seems to be of minor importance. Therefore, as we proposed in our paper, galanin can be engaged in the transduction of special kind of sensation like dyspepsia, as a defense mechanism against consuming too much food in ulcer disease. 

However, as the Reviewer correctly noted, according to many studies galanin acts as a modulator of inflammatory reaction, and its increasing production is considered to be aimed at limiting acute-phase inflammatory responses to prevent excessive inflammation and to restore homeostasis (PMID: 21087790). Such galanin action is mediated by reduction of tissue sensitivity to neurogenic inflammation factors (as SP, CGRP) or regulation of proinflammatory cytokine production (PMID: 21087790). Gastric ulcer model examined in our study was accompanied by a strong inflammatory response, thus galaninergic nerve response may be additionally associated with the regulation of the inflammatory process itself. Therefore, increased number of Galanin-immunofluorescent perikarya supplying the ulcered stomach, observed in both, the gastric wall (PMID: 29717796, PMID: 27175780 ) and nodose ganglia (present article, PMID: 26917098), suggests such inflammation modulating action of galanin in ulcer disease. Since GalR3 mediates antiedema effect in the skin (PMID: 18679831), its increased expression observed in our study ameliorates the thesis on the galanin mediated inhibition of plasma extravasation in ulcered tissue and inhibiting neurogenic inflammation. The hypothesis on anti-inflammatory function of galanin in acute ulcer disease is strongly supported by experiments demonstrating its protective and anti-inflammatory effects in acute inflammatory phase of experimentally induced colitis (PMID: 16846834, PMID: 17331599). 

Such paragraph has been incorporated into the Discussion.

7. The manuscript should be edited for grammar. 

The manuscript has been verified by a native speaker.

---

## [Editor Report · Decision Letter 2]

9 Nov 2020

Inferior vagal ganglion galaninergic response to gastric ulcers

PONE-D-20-20776R2

Dear Dr. Zalecki,

We’re pleased to inform you that your manuscript has been judged scientifically suitable for publication and will be formally accepted for publication once it meets all outstanding technical requirements.

Kind regards,

Tudor C Badea, M.D., M.A., Ph.D.

Academic Editor

PLOS ONE
---

## [Editor Report · Acceptance letter]

12 Nov 2020

PONE-D-20-20776R2 

Inferior vagal ganglion galaninergic response to gastric ulcers 

Dear Dr. Zalecki:

I'm pleased to inform you that your manuscript has been deemed suitable for publication in PLOS ONE. Congratulations! Your manuscript is now with our production department. 

Kind regards, 

on behalf of

Dr. Tudor C Badea 

Academic Editor

PLOS ONE